# Effects of High Cervical Spinal Cord Stimulation on Gait Disturbance and Dysarthropneumophonia in Parkinson’s Disease and Parkinson Variant of Multiple System Atrophy: A Case Series

**DOI:** 10.3390/brainsci12091222

**Published:** 2022-09-10

**Authors:** Linbin Wang, Rui Zhu, Yixin Pan, Peng Huang, Yuyan Tan, Boyan Fang, Jun Liu, Dianyou Li

**Affiliations:** 1Institute of Science and Technology for Brain-Inspired Intelligence (ISTBI), Fudan University, Shanghai 200000, China; 2Department of Neurosurgery, Center for Functional Neurosurgery, Ruijin Hospital, Shanghai Jiao Tong University School of Medicine, Shanghai 200000, China; 3Department of Neurology, Ruijin Hospital, Shanghai Jiao Tong University School of Medicine, Shanghai 200000, China; 4Neurological Rehabilitation Center, Beijing Rehabilitation Hospital, The Affiliated Hospital of Capital Medical University, Badachu Road, Shijingshan District, Beijing 100000, China

**Keywords:** gait disturbance, dysarthria, Parkinson’s disease, multiple system atrophy, spinal cord stimulation

## Abstract

High cervical spinal cord stimulation (HCSCS) was found to have therapeutic effects on Parkinsonian gait disturbance. However, the results were inconsistent and confounded with symptoms of pain. This study aimed to reveal the gait and dysarthric effects of HCSCS in PD (Parkinson’s disease) and MSA-P (Parkinson variant of multiple system atrophy) patients without pain. Three PD and five MSA-P patients without painful comorbidities were assessed for gait performance and speech before SCS surgery and at 3- and 6-month follow-up. Stride length and the time spent in the Timed Up-and-Go task showed little change after HCSCS surgery. Overall voice quality (measured by dysphonia severity index) and perceptual speech intelligence improved significantly at 3 months, but improvements slightly diminished at 6 months postoperatively. Change in quality of life (measured by 8-item Parkinson’s disease questionnaire) was also notable at 3 months but narrowed over time following HCSCS. In conclusion, HCSCS showed therapeutic effects in improving the dysarthria but not gait disturbance in pain-free PD and MSA-P patients.

## 1. Introduction

Parkinson’s disease (PD) and Parkinsonian variant of multiple system atrophy (MSA-P) are characterized by dominant parkinsonian symptoms and share a common pathological abnormality in the nigrostriatal regions [1,2,3]. The majority of patients with PD and MSA-P could develop disabling symptoms, including speech and swallow disorders, postural instability, and gait problem that are refractory to medication and deep brain stimulation (DBS) in the advanced stage, negatively influencing patients’ quality of life.

Spinal cord stimulation (SCS) is a well-established operative treatment for the management of intractable neuropathic pain [4]. SCS also has therapeutic effects on motor deficits assumably by modulating the ascending sensory pathway and disrupting the abnormal low-frequency cortical-striatal oscillation [5,6,7,8].

Dorsal column spinal stimulation at high cervical levels was introduced as one of the SCS modalities, with superiority in treating pain from associated dermatomes [9]. A positive effect of high cervical SCS (HCSCS) on motor symptoms has been reported in idiopathic PD and atypical Parkinsonism, yet often confounded with the postoperative pain relief [10,11,12].

To this end, we reviewed 3 PD patients and 5 MSA-P patients who had no comorbidity of pain. The primary goal of this study was to determine the effects of HCSCS on gait disturbance in patients with PD and MSA-P. Considering the common pathology between gait disorders and speech disturbance and the functional role of the cervical spinal cord in speech production, we also speculated that HCSCS alleviates dysarthria in PD and MSA-P. Hence, the patients also received quantitative acoustic and perceptual dysarthria measurements that range over the multidimensional aspects of speech before and after SCS surgery, to reveal the dysarthric effects of HCSCS.

## 2. Materials and Methods

### 2.1. Participants

This study was approved by the ethics committee of Ruijin Hospital and registered at Chinese Clinical Trial Registry (ChiCTR1900024326). All patients signed written informed consent.

We retrospectively reviewed 3 patients with PD and 5 with MSA-P who received HCSCS for treating medication-refractory postural instability and gait difficulties from June 2020 to June 2021, at the Department of Functional Neurosurgery, Ruijin Hospital, Shanghai Jiao Tong University School of Medicine. The inclusion criteria include: (1) established diagnosis of primary PD (based on the UK Parkinson’s Disease Society Bank Criteria) or MSA-P (according to consensus diagnostic criteria), (2) absence of painful comorbidity other than dystonia and prior trauma [13], and (3) completion of baseline and 6-month follow-up evaluations.

### 2.2. Outcome Measurements

All patients received gait assessment and speech examination preoperatively in the off-medication state (i.e., medication withdrawal overnight or more than 12 h). Three and six months after the SCS was turned on, they were reassessed in the off-medication state.

Gait performance was measured using a 5-m timed up and go (TUG) test. The time spent in TUG and the stride length were obtained.

Speech recordings were performed with patients sitting in a quiet room and adapted to a head-mounted condenser microphone (Atmos, Germany). Speech signals were sampled at 22.4 kHz with 16-bit resolution, and preprocessed using LingWAVES software. The acoustic measures associated with the speech subsystems of articulation, phonation, prosody, resonance, and respiration were obtained, including the median fundamental frequency (f0) and its standard deviation, the median relative intensity, jitter, shimmer, harmonic-to-noise ratio (HNR), maximum phonation time (MPT) and vowel space area (VSA) [14]. A multiparametric dysphonia severity index (DSI) was calculated to indicate the acoustic voice quality [15]. The perceptual speech intelligence was estimated using the speech item of the MDS UPDRS-III.

The following were also measured at 3 and 6 months and compared to baseline: part III of the Movement Disorder Society-Unified Parkinson’s Disease Rating Scale (MDS UPDRS-III) and its rigidity (item 3), bradykinesia (item 4–8, 14), axial (item 1–2, 9–13) subscores, and 8-item Parkinson’s disease questionnaire (PDQ-8).

### 2.3. Surgical Procedure

Surgery was performed in the Department of Functional Neurosurgery at the Ruijin Hospital in the period from June 2020 to June 2021.

We employed a modified retrograde surgical lead insertion technique for lead implantation. Briefly, patients under general anesthesia were placed prone with head stabilized by the Mayfield frame. Two partial laminectomies were performed at the level of C2 and C3, respectively. In a few cases, three-level partial laminectomies were performed due to difficulty in lead insertion or to decompression purpose. The surgical plate electrode (2 × 8 contacts, model 39286, Medtronic, Minneapolis, MN, USA) was then inserted downward from the C2 laminectomy window at the midline of the dura at C2 and C5. The electrode orientation could be directly visualized and adjusted from the lower-level laminectomy window in reference to motor responses during the intraoperative testing. After the fixation, the lead was temporarily connected to an external stimulator (model 355531, Medtronic, Minneapolis, MN, USA) by extension wire for trial stimulations during several days.

The suprathreshold tonic stimulation was applied and stimulation parameters were fine-tuned with well-tolerated paresthesia. If the clinical response (A sign of gait improvement presented and no unwanted effects occurred) in the test phase (less than 4 weeks) was satisfactory, the patient underwent the second-stage surgery involving the implantation of the SCS neurostimulator (model 37714, Medtronic, Minneapolis, MN, USA) in the subclavicular fossa. The electrode locations were verified by postoperative X-ray and the pair of stimulation contacts with optimal combination of stimulation parameters were individually determined (Figure 1).

### 2.4. Statistical Analyses

Kolmogorov-Smirnov test was applied to test the normality of data distribution. Preoperative baseline scores were then compared with scores at 3, 6 months postoperatively by use of the Wilcoxon paired-rank test or paired-sample t test for significance.

The statistical analyses combined PD and MSA-P group due to the small sample size for each group, whereas descriptive statistics of the two groups were separated out in the tables and figures to provide detailed information.

All analyses were performed using SPSS 20.0. *p* < 0.05 was considered significant.

## 3. Results

### 3.1. Demographic Data and Stimulation Parameters

Three patients with PD (3 males, mean age 64 ± 11 years, disease duration 8 ± 5 years) and five patients with MSA-P (2 males and 3 females, mean age 60 ± 5 years, disease duration 4 ± 2 years) participated in the study. Among them, one PD patient and two MSA-P patients had been previously treated with STN-DBS. No complications occurred during electrode implantation and in the follow-up period. Other demographic data are shown in Table 1.

The stimulation was initially set at frequency range of 10–30 Hz, pulse-width range of 90–210 μs, and stimulation-amplitude range of 0.5–2.0 V. Gait speed and stride length were typically used to assess the response to HCSCS. At each follow-up, acute stimulation effects of 25 Hz, 45 Hz, 60 Hz, and 130 Hz were tested separately with a fixed pulse-width at 210 μs, and then parameters were fine-tuned to determine the optimal combination of frequency and pulse width. The threshold and therapeutic stimulation parameters in response to postural changes for each patient were established using automatic position-adaptive stimulation. Stimulation parameters determined for each patient over different follow-up periods are shown in Table 2. Notably, the final stimulation frequencies chosen for the patients ranged from 15 to 60 Hz (Appendix A).

### 3.2. Main Results

For TUG task, comparison of SCS preoperative versus postoperative states did not reveal significant influence on stride length (PD: unchanged at 3-month and 20% improvement at 6-month; MSA-P: 8% decline at 3-month and 10% decline at 6-month) and the time spent in TUG (PD: 15% decline at 3-month and 14% decline at 6-month; MSA-P: 40% improvement at 3-month and 28% improvement at 6-month) (Figure 2), although two out of eight patients got gait improvements. Compared to the baseline, total MDS UPDRS-III score and its rigidity, bradykinesia and axial subscales remained unchanged postoperatively (Table 3).

In terms of the acoustic parameters, DSI score decreased significantly at both 3-month (PD: increased by 4.3; MSA-P: increased by 4.9) and 6-month follow-up visits (PD: increased by 5.8; MSA-P: increased by 4.4) when compared to the baseline. Among them, six patients at 3-month and five patients at 6-month exceeded 2.49 in DSI changes. Particularly, the benefit declined at 6-month in MSA-P patients but remained generally stable in PD patients. Meanwhile, perceptual speech intelligence as measured by speech item of the MDS UPDRS-III showed a similar pattern: the symptom score on speech reduced remarkably at 3-month (PD: 37% reduction; MSA-P: 27% reduction) but slightly restored at 6-month (PD: 25% reduction; MSA-P: 27% reduction) postoperatively (Figure 2).

The median intensity was also remarkably increased at 3-month and 6-month postoperative periods, whereas the f0 made no change. Jitter, shimmer and HNR showed signs of improvement after SCS implantation, and all reached a significant level at 6-month follow-up. The standard deviation of f0, however, worsened during 3-month follow-up but partially restored at 6-month postoperative period. Meanwhile, MPT and VSA remained unchanged after surgery (Table 4).

Similarly, change in life of quality as measured by PDQ-8 score was notable at 3 months, but narrowed almost to vanishing point at 6 months following HCSCS (Table 3).

## 4. Discussion

In this pilot study, we investigated the effect of HCSCS on gait disturbance and dysarthria in pain-free PD and MSA-P patients, comparing HCSCS preoperative and postoperative treatment states when patients were off-medication. Collectively, our results showed positive results in speech intelligence and voice quality, but not in gait performance. This is further supported by the observed improvements in articulatory and phonatory parameters. Following the improvement in dysarthria, quality of life also improved in a similar way.

In previous studies with HCSCS, except for the initial exploratory work on two patients at a ten-day postoperative phase, two other studies reported delayed motor benefits induced by HCSCS, which required three months of application to take effects [10,11,12]. Different from these findings, no effect on gait or motor function from HCSCS was observed even at 6-month in our study. The first explanation for the discrepancy is that our patients have no painful symptoms, and thus a beneficial effect on PD symptoms from the relief of pain is minimized in our patients [12,16,17,18]. Secondly, MSA-P progresses more rapidly than PD. The mixed patient cohort of PD and MSA-P could undermine the potential therapeutic effect. Moreover, among these patients with PD or MSA-P, the disease duration varied considerably between PD and MSA-P and also from person to person. Early-stage patients with good levodopa responsiveness, that rarely involve extranigral nondopaminergic abnormalities, may benefit more from HCSCS as suggested from previous studies [19]. Thirdly, the individually optimized albeit heterogeneous stimulation protocols of our study may also account for heterogeneous results in previous studies [20]. In two previous studies with positive results, one selected 40 Hz [12] and the other selected 135–185 Hz [10], whereas the stimulation frequencies ultimately chosen for our patients ranged from 15 to 60 Hz.

Similar to the effect on Parkinsonian speech of subthalamic stimulation [21], the dysarthric improvement is observed following HCSCS, as better scores of DSI and speech item of the MDS UPDRS-III. DSI score has been suggested as a stable and sensitive index to describe differences in vocal capability between treatment conditions [22,23]. Furthermore, the indices of the acoustic analysis revealed improvements in two of the main components of speech production: articulation (intensity) and phonation (jitter, shimmer and HNR). Specific deficits in these two components are also regarded as the main cause of Parkinsonian hypokinetic dysarthria [24]. Previous studies suggested HCSCS could enhance the endogenous sensory input by stimulating the myelinated fibers in the dorsal horn of the spinal cord, thereby facilitating the sensorimotor integration in speech production [18]. In addition, HCSCS may promote top-down control over speech production by modulating the ascending sensory pathways, activating the brain areas that overlap with the language networks, and introducing a more goal-directed articulatory pattern [25]. Furthermore, our results underscore the postulation that HCSCS might be superior in recruiting brainstem arousal systems and affecting the innervated speech and swallow function, rather than lower extremity motor function [20].

The study has several limitations. First, the sample size is small, which may lead to a higher rate of type II error, and thus the differences between MSA-P and PD are unable to be compared either. Secondly, since SCS causes obvious paresthesia, the placebo effect may be inevitable in an uncontrolled open-label design. Thirdly, the disease duration was not homogeneous among the patients. The efficacy of HCSCS in patients with different disease duration may vary. The relationship between the disease duration and the effectiveness of HCSCS warrants further investigation with a cohort of a larger sample size. Finally, dysarthria and oropharyngeal dysphagia are closely related, and share overlapped pathological mechanisms [26], and thus improvement in swallowing is also expected after HCSCS. However, a systematic evaluation of swallow function is absent in our study.

## 5. Conclusions

HCSCS showed therapeutic effects in improving the dysarthria but not gait disturbance of pain-free PD and MSA-P patients. However, long-term effects need to be verified in a large-sample, prospective, randomized controlled trial.

## Figures and Tables

**Figure 1 brainsci-12-01222-f001:**
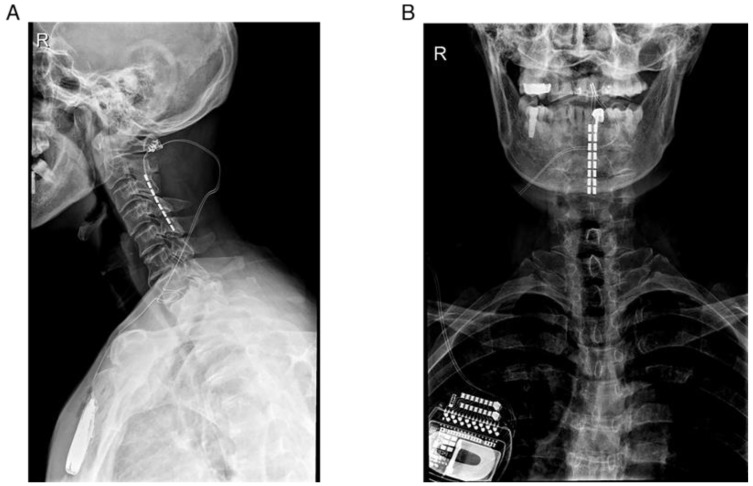
Representative example showing the correct positioning of the octopolar electrode at C2-C5 level by postoperative radiography. (**A**) Medial-lateral plane, (**B**) Anterior-posterior plane.

**Figure 2 brainsci-12-01222-f002:**
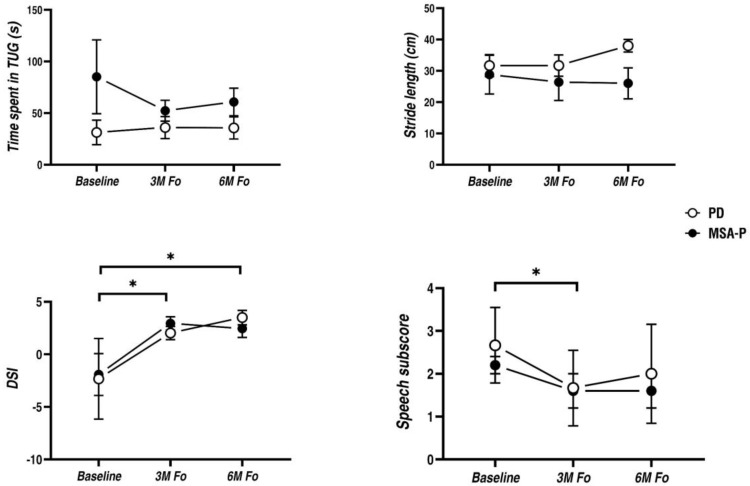
Effects of HCSCS on gait and dysarthria in PD and MSA-P patients at 3-month and 6-month postoperative treatment states when patients were off-medication. Upper left: time to complete TUG task; Upper right: stride length; Bottom left: DSI score; Bottom right: speech intelligence, as measured by the speech item of MDS UPDRS-III. Means are plotted with error bars representing the standard errors of the mean. PD and MSA-P data are plotted separately by hollow circle and solid circle. * *p* < 0.05 indicates significant difference between conditions.

**Table 1 brainsci-12-01222-t001:** Demographic and clinical characteristics.

Patient	Age (Year)	Gender	LEDD (mg)	Diagnosis	Disease Duration (Year)	DBS	Lead Location	Follow Up Period (mo)
No.1	53	M	1164.25	PD	13	Yes	C2–C5	6
No.2	75	M	550	PD	7	No	C2–C5	6
No.3	63	M	900	PD	4	No	C2–C5	6
No.4	60	F	600	MSA-P	6	No	C2–C5	6
No.5	61	F	500	MSA-P	6	No	C2–C5	6
No.6	58	F	575	MSA-P	4	Yes	C2–C5	6
No.7	54	M	300	MSA-P	1	No	C2–C5	7
No.8	67	F	(-)	MSA-P	3	No	C2–C5	7

(-) indicated unmedicated patient. Abbreviations: F, female; M, male; LEED, levodopa equivalent daily dose; mo, months; PD, Parkinson’s disease; MSA-P, Parkinson Variant of Multiple System Atrophy.

**Table 2 brainsci-12-01222-t002:** Changes in stimulation parameters over different follow-up periods.

Patient	Diagnosis	3M Fo			6M Fo		
		Frequency (Hz)	Pulse Width (μs) ^#^	Amplitude (V) ^#^	Frequency (Hz)	Pulse Width (μs) ^#^	Amplitude (V) ^#^
No.1	PD	20	120	1.60	20	150	1
No.2	PD	35	280	2	40	280	2.5
No.3	PD	30	240	2	30	135	2
No.4	MSA-P	15	220	0.45	15	120	0.6
No.5	MSA-P	25	210	1.15	45	210	1.5
No.6	MSA-P	45	310	0.8	35	180	0.55
No.7	MSA-P	20	240	1.8	30	260	3.8
No.8	MSA-P	25	240	0.8	25	240	1.5

^#^ Mean therapeutic amplitude (V) and pulse width (μs) in the upright body position. Abbreviations: 3M Fo, 3-month follow-up; 6M Fo, 6-month follow-up; PD, Parkinson’s disease; MSA-P, Parkinson Variant of Multiple System Atrophy.

**Table 3 brainsci-12-01222-t003:** Motor symptoms and life of quality.

		Baseline	3M Fo	6M Fo	*p* Value (3M Fo vs. Baseline)	*p* Value (6M Fo vs. Baseline)
**MDS UPDRS-III**						
Total	PD	44 (5)	39 (3)	37 (1)	0.268	0.121
	MSA-P	44 (16)	44 (14)	44 (18)		
Rigidity subscore	PD	8 (5)	5 (2)	2 (2)	0.134	0.129
	MSA-P	9 (4)	8 (4)	9 (3)		
Bradykinesia subscore	PD	21 (1)	22 (1)	21 (2)	0.154	0.756
	MSA-P	21 (14)	23 (12)	21 (14)		
Axial subscore	PD	13 (3)	12 (1)	12 (4)	0.133	0.476
	MSA-P	14 (4)	13 (4)	14 (4)		
**PDQ-8**						
Total	PD	10 (2)	5 (4)	9 (7)	<0.05	0.397
	MSA-P	14 (2)	10 (3)	12 (2)		

Mean ± SD values of severity of motor symptoms (MDS UPDRS-III) as well as life of quality (PDQ-8) separately in the 3 PD patients and in the 5 MSA-P patients. Abbreviations: 3M Fo, 3-month follow-up; 6M Fo, 6-month follow-up; PD, Parkinson’s disease; MSA-P, Parkinson Variant of Multiple System Atrophy; MDS UPDRS-III, the part III of Movement Disorder Society-Unified Parkinson’s Disease Rating Scale; PDQ-8, eight-item Parkinson’s disease questionnaire.

**Table 4 brainsci-12-01222-t004:** Acoustic parameters.

		Baseline	3M Fo	6M Fo	*p* Value (3m Fo vs. Baseline)	*p* Value (6m Fo vs. Baseline)
**Articulat** **ion**						
*f0* (%)	PD	152 (14)	148 (32)	148 (30)	0.401	0.123
	MSA-P	171 (37)	189 (53)	226 (82)		
Intensity (dB)	PD	67 (14)	78 (4)	81 (1)	<0.01	<0.05
	MSA-P	68 (5)	77 (6)	76 (4)		
**Phonation**						
Jitter (%)	PD	2.9 (3.8)	0.3 (0.1)	0.2 (0.1)	<0.05	<0.05
	MSA-P	3.6 (2.9)	0.3 (0.1)	0.2 (0.2)		
Shimmer (%)	PD	17.2 (6.7)	11.9 (2.5)	12.1 (3.6)	0.073	<0.05
	MSA-P	22.8 (15.6)	9.5 (3.1)	7.4 (3.0)		
HNR (dB)	PD	4.6 (3.6)	9.0 (1.9)	11.0 (2.3)	0.236	<0.05
	MSA-P	11.5 (4.4)	11.5 (5.1)	16.8 (5.3)		
**Resonance**						
VSA (Hz^2^)	PD	72,608 (29,029)	282,290 (147,647)	80,814 (59,988)	0.069	0.401
	MSA-P	111,526 (161,014)	175,379 (87,519)	156,835 (79,689)		
**Respiration**						
MPT (s)	PD	8.8 (5.3)	8.9 (3.4)	13.0 (8.0)	0.582	0.401
	MSA-P	7.9 (3.9)	6.8 (3.9)	8.7 (7.4)		
**Prosody**						
SD *f0* (%)	PD	2.3 (3.0)	1.2 (0.8)	2.0 (2.7)	<0.05	0.069
	MSA-P	19.5 (11.0)	1.3 (1.0)	4.6 (8.1)		

Mean ± SD values of acoustic parameters (including MPT, Jitter, Shimmer, *f0*, and SD *f0)* separately in the 3 PD patients and in the 5 MSA-P patients. Abbreviations: 3M Fo, 3-month follow-up; 6M Fo, 6-month follow-up; PD, Parkinson’s disease; MSA-P, Parkinson Variant of Multiple System Atrophy; *f0*, fundamental frequencies; HNR, harmonic-to-noise ratio; VSA, vowel space area; MPT, Maximal phonatory time.

## Data Availability

Not applicable.

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
