# Peer review of "Effects of High Cervical Spinal Cord Stimulation on Gait Disturbance and Dysarthropneumophonia in Parkinson’s Disease and Parkinson Variant of Multiple System Atrophy: A Case Series"

_brainsci, 2022, doi:10.3390/brainsci12091222_

Round 1

Reviewer 1 Report (Previous Reviewer 2)

The authors have improved the manuscript. However, I have still some minor comments to the authors:

-       In the methods the authors should better specify that dystonia and prior trauma were excluded among the painful condition (point 2 of the inclusion criteria). I suggest including this reference that can help the reader to better understand the topic:

Defazio G, et al. Does acute peripheral trauma contribute to idiopathic adult-onset dystonia? Parkinsonism Relat Disord. 2020

-       The authors should also include among the limitations that disease duration was not homogeneous among the patients included in the study.

Author Response

Reviewer 2 Report (Previous Reviewer 1)

Much improved manuscript, with only the following minor suggestions.

Line150-151 Recommend replacing "worsening" with decline 

Line 197 should "expect" be "except"

Line 208 Recommend replacing "inspired" with "suggested"

Author Response

This manuscript is a resubmission of an earlier submission. The following is a list of the peer review reports and author responses from that submission.

Round 1

Reviewer 1 Report

Abstract-Please define all abbreviations including  MSA-P, SCS, DSI

The introduction could include greater background on HCSCS effects on motor symptoms and/or function.  In line 289, authors mention that their findings differ from previous findings, but little is mentioned in the introduction regarding previous motor findings of HCSCS related to PD, MSA-P, or other motor conditions.

Methods section needs to include the stimulation frequency used and either in the Methods or Discussion the rationale for selecting the stimulation frequency of 15-60 HZ when other studies used 40 Hz and 135-185 Hz.

Line 289 needs citations provided regarding “previous studies”

Lines 154-156. Please clarify the description of the f0 standard deviation worsened during the 3 month follow-up (agreed) but quickly restored at 6 month postoperative period (this doesn’t appear correct based on Table 4 standard deviations which appear much larger than at baseline). Please check/clarify.

The results are provided with percent improvement or percent reduction, without specifying whether it was the PD group or MSA-P group (or combined). The tables separate out the PD and MSA-P, but there is no percent change provided in the manuscript text for each group. Figure 2 splits out the two groups, so it is unclear for example how the median intensity increased 13.2% and yet the groups were separated out in the tables  (is this for all 8 participants?) Please clarify.

Discussion-among those with PD or MSA-P, did the duration of the condition 4yrs vs 13yrs in PD or 1 vs6 years in MSA-P impact the outcomes? While it may not be able to determine this statistically due to the small sample size, it might be informative if the authors share some thoughts regarding impact of the disease duration on HCSCS-related outcomes.

Reviewer 2 Report

The authors reported a study on the effects of high cervical spinal cord stimulation in Parkinson’s Disease and Parkinson Variant of Multiple System Atrophy. I have some comments to the authors:

-       The title should reflect that the study is a case series (only 8 patients are included in this paper and no control group)

-       In the methods the authors should state how they assessed rigidity/Bradikynesia/axial subscores

-       The authors should assess the p value for each group alone (e.g. PD patients and MSA-P patients separately).

-       Did the authors compare these findings with a control population of PD patients and MSA-P patients respectively? 

-       Table 3 is not clear, it seems that there were extra lines.